# Structural basis of oncogenic histone H3K27M inhibition of human polycomb repressive complex 2

Neil Justin[1,*], Ying Zhang[1,*], Cataldo Tarricone[1,*], Stephen R. Martin[2], Shuyang Chen[1], Elizabeth Underwood[1], Valeria De Marco[1], Lesley F. Haire[1,2], Philip A. Walker[2], Danny Reinberg[3], Jon R. Wilson[1] & Steven J. Gamblin[1]

Polycomb repressive complex 2 (PRC2) silences gene expression through trimethylation of K27 of histone H3 (H3K27me3) via its catalytic SET domain. A missense mutation in the substrate of PRC2, histone H3K27M, is associated with certain pediatric brain cancers and is linked to a global decrease of H3K27me3 in the affected cells thought to be mediated by inhibition of PRC2 activity. We present here the crystal structure of human PRC2 in complex with the inhibitory H3K27M peptide bound to the active site of the SET domain, with the methionine residue located in the pocket that normally accommodates the target lysine residue. The structure and binding studies suggest a mechanism for the oncogenic inhibition of H3K27M. The structure also reveals how binding of repressive marks, like H3K27me3, to the EED subunit of the complex leads to enhancement of the catalytic efficiency of the SET domain and thus the propagation of this repressive histone modification.

[1] The Francis Crick Institute, Mill Hill Laboratory, London NW7 1AA, UK. [2] Structural Biology Science Technology Platform, Francis Crick Institute, Mill Hill Laboratory, London NW7 1AA, UK. [3] Department of Molecular Pharmacology and Biochemistry, New York University School of Medicine, New York, New York 10016, USA. * These authors contributed equally to this work. Correspondence and requests for materials should be addressed to S.J.G. (email: steve.gamblin@crick.ac.uk).

Epigenetic mechanisms play an important role in setting and maintaining gene expression profiles that define the fate of a cell[1]. The polycomb group of proteins (PcG) are multi-subunit complexes that maintain repressive chromatin states by silencing the expression of certain genes[2,3]. PRC2 mediates this function by methylation of lysine 27 of histone H3, a hallmark of repressive chromatin[4]. Three core subunits make up the catalytic core of PRC2; the SET domain containing EZH2, the zinc-finger containing SUZ12 and the WD40 repeat protein EED. We have previously shown that the EED subunit of PRC2 recognises repressive marks, such as H3K27me3, the product of PRC2 activity, leading to the allosteric activation of the lysine methyltransferase activity of the complex[5]. Thus H3K27me3 functions as a repressive mark in terms of gene expression but has an activating capacity with respect to PRC2 function. Further, that ablation of this EED binding activity leads to global reduction in histone methylation and disruption of *Drosophila* development. A second WD40 repeat protein RbAp46/48 binds to this catalytic core to target nucleosomes[6], as do additional regulatory components, that function in certain contexts, including Aebp2, PHF1 and Jarid2 (refs 7–9). We have shown that Jarid2 is trimethylated by PRC2 leading to its binding to the EED subunit and stimulating PRC2 in a manner analogous to activation by H3K27me3 (ref. 10). Jarid2 methylation promotes PRC2 activity at a locus lacking H3K27me3, ensuring the correct setting of this mark for gene expression.

In addition to crystallographic data on peptide binding by EED, important insights into the PRC2 mechanism have come from a number of structures of other components of the complex. Structures of RbAp48 subunit, and homologues, in complex with a SUZ12 peptide and a histone H3 peptide helped explain PRC2 recognition of its nucleosomal substrate[11,12]. The EZH2 SET domain structure revealed an auto-inhibited conformation in the absence of the other PRC2 subunits, suggesting their requirement for activity as well as regulation[13]. Finally, a highly informative negative-stain electron microscopy study provided the first description of the overall architecture of human PRC2 complex (EZH2/EED/SUZ12/RbBP4/AEBP2)[14].

The gain of function mutations in the SET domain of EZH2 were found to be associated with a subtype of B-cell lymphomas, follicular lymphoma and melanoma[15,16]. Recently, another gain of function mutation but this time in the histone H3 substrate of PRC2, H3K27M, was identified in pediatric glioblastoma, functioning as a dominant negative, resulting in a drastic decrease of H3K27me3 in the affected cells[17,18]. This effect was shown to arise from the mutant histone interacting with the EZH2 subunit of PRC2 and inhibiting its catalytic activity[19]. Since the mutation is present in one of the many histone H3.1, or few H3.3, how this limiting mutant polypeptide has such a dominant effect remains unclear.

A recent landmark achievement was the report of a crystal structure of an active fragment of PRC2 (ref. 20). The structure, from *Chaetomium thermophilum,* has been reported in both an activated and unactivated conformation with a peptide representing H3K27M bound to the SET domain. Here we present the crystal structure of the human PRC2 with bound H3K27M peptide that, in contrast to the *C. thermophilum* structure, shows the substituted methionine residue occupying the lysine access channel of the catalytic SET domain supporting a model of inhibition consistent with the known biochemical data. The structure also suggests a mechanism for the allosteric activation of PRC2 by neighbouring repressive marks.

## Results

We have solved the crystal structure of a human PRC2 complex that contains the essential elements of the EZH2 and EED polypeptides with the Vefs fragment of the SUZ12 subunit (Fig. 1). In addition, the complex contains an SAH cofactor and two biologically relevant peptides; the first from Jarid2 containing K116me3, which is a high affinity activator that acts in the same manner as H3K27me3, and is bound to the EED domain. The second, based on the oncogenic histone H3 mutation H3K27M, binds to the SET domain and inhibits its methyltransferase activity. The crystal structure was solved using a combination of phases from zinc anomalous scattering and molecular replacement (see Methods section). There were four copies of the PRC2 complex in the crystallographic asymmetric unit (described in Supplementary Fig. 1) and the final model was refined at 2.9 Å resolution, and relevant crystallographic statistics are given in Table 1. During the course of building the human complex, coordinates became available for a related complex from the thermophilic fungus *C. thermophilum*, which were subsequently used to guide final building. For clarity, in our description of the human structure, we have adopted, where appropriate, the domain nomenclature used in the earlier paper. A detailed comparison between the human and *C. thermophilum* PRC2 structures is presented in Supplementary Fig. 2.

**Overall architecture of the human PRC2 complex.** The complex forms a compact arrangement of three lobes with overall dimensions of approximately $120 \times 75 \times 50$ Å (Fig. 1c). The first two lobes are involved in regulation and the third in catalysis. The core of the N lobe is formed from the seven-bladed β-propeller structure of EED and around this wraps the amino (N)-terminal region of EZH2 (residues 10 to 246), which is made up of six distinct subdomains, the first and last of which interact. The middle lobe largely comprises two domains that mark the beginning of the carboxy (C)-terminal region of EZH2 (MCSS and SANT2) and the helical, C-terminal, component of the Suz12 Vefs domain. The third Catalytic lobe mostly comprises the pre-SET and SET domains from the C terminus of EZH2 but also includes the N terminus of the Vefs domain.

*N lobe.* The first subdomain of EZH2, termed SBD, forms a long helix that packs against the SANT1 domain, the final subdomain of the EZH2 N-terminal fragment, completing a four-helix bundle. Thus, the first and last of the domains from the EZH2 (N-domain) interact to close the 'lasso' that EZH2 makes around the EED subunit. The EBD and BAM domains, helical and β-structures respectively, pack against the bottom and side of EED and are currently of unknown function. The function of the next two elements of EZH2, referred to as SAL and SRM, is clearer. Together they link one side of the SET domain to the EED subunit and communicate the binding of repressive trimethyl signals to EED to the catalytic centre of the SET domain, stimulating activity. The detailed mechanism of this process is discussed below. The SAL chain runs between the SET, EED and Vefs domains before switching back, passing behind the SET-I helix and forming an extended chain and helix (SRM) that sits across the top 'open end' of the EED barrel making extensive interactions. The SRM helical segment runs adjacent to the Jarid2 peptide at the top of EED with the main chains about 8 Å apart and making extensive side-chain interactions between the two segments. Similarly, most of the contacts made by the SRM helix with the SET-I helix are mediated by side chains.

*Middle lobe.* The middle lobe probably has regulatory and targeting functions. For example, a helical segment (residues 330–343) at the C-terminal of the SANT2 domain is known to be involved in RNA binding[21]. There are multiple contacts between the EZH2

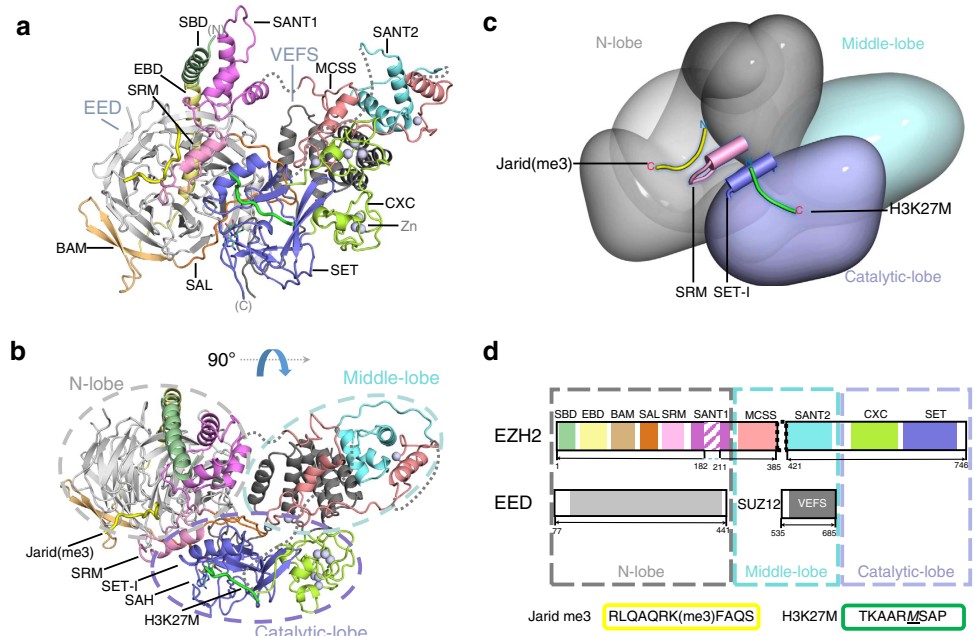

**Figure 1 | Overall structure of the human PRC2 complex.** (**a**) Structure of the catalytically active core of PRC2 in cartoon representation. (**b**) Orthogonal view. (**c**) Stylized surface representation showing the relationship of the three lobes of PRC2. The N lobe consists of EED and the N-terminal region of EZH2. The middle lobe of the SUZ12 Vefs and EZH2 MCSS and SANT2. The catalytic lobe comprises the post SET and SET domains and a hairpin formed by the region of Suz12 N-terminal to the Vefs. Superposed is a cartoon indicating the activating Jarid2 peptide (yellow), which stabilizes the SRM region (pink), which in turn stabilizes the SET-I helix (blue) and thus completes the active site where substrate peptide (green) is bound. (**d**) Schematic representation of the domain architecture of PRC2 showing the composition of the complex. The distinct subdomains discussed in the main text and shown in **a,b** are indicated by the coloured regions.

**Table 1 | Data collection and refinement statistics.**

|  | Native (5HYN) | Zn Data set |
|---|---|---|
| *Data collection* |  |  |
| Wavelength (Å) | 0.9763 | 1.2826 |
| Space group | $P2_12_12_1$ | $P2_12_12_1$ |
|  |  |  |
| Cell dimensions |  |  |
| *a, b, c* (Å) | 131.64, 171.51, 274.55 | 132.00, 173.08, 274.96 |
| *α, β, γ* (°) | 90.00, 90.00, 90.00 | 90.00, 90.00, 90.00 |
| Resolution (Å) | 104.43 − 2.94 (3.02 − 2.94) | 119.00 − 3.19 (3.27 − 3.19) |
| $R_{sym}$ or $R_{merge}$ | 0.16 (1.61) | 0.42 (3.28) |
| $I/\sigma I$ | 9.9 (1.4) | 8.5 (1.3) |
| $CC_{1/2}$ (%) | 99.6 (45.8) | 99.7 (47.0) |
| Completeness (%) | 99.9 (99.9) | 99.5 (99.1) |
| Redundancy | 5.6 (5.6) | 19.4 (20.2) |
|  |  |  |
| *Refinement* |  |  |
| Resolution (Å) | 104.43 − 2.95 |  |
| No. of reflections | 130,834 |  |
| $R_{work}/R_{free}$ | 0.22/0.27 |  |
| No. of atoms |  |  |
| Protein | 34,297 |  |
| Ligand (H3K27M) | 252 |  |
| Ligand (JARIDme3) | 343 |  |
| Ligand (SAH) | 104 |  |
| Ligand (Zn) | 32 |  |
| *B*-factors |  |  |
| Protein | 88.73 |  |
| Ligand (H3K27M) | 83.86 |  |
| Ligand (JARIDme3) | 94.55 |  |
| Ligand (SAH) | 84.57 |  |
| Ligand (Zn) | 67.64 |  |
| R.m.s. deviations |  |  |
| Bond lengths (Å) | 0.013 |  |
| Bond angles (°) | 1.523 |  |

MCSS, the SANT2 domain and the Vefs domain. This integrates the MCSS domain from the end of the EZH2$^N$ construct and the SANT2 from the beginning of the EZH2$^C$ construct into an extended helical domain. There are two zinc-binding motifs in the middle lobe; one ($Zn_1Cys_3His_1$) is formed solely by MCSS and the second ($Zn_1Cys_4$) by two cysteines contributed by SANT2 and two from the MCSS domain.

*Catalytic lobe.* The C terminus of EZH2 contains the pre-SET and SET domains; the former is characterized by two zinc-binding motifs, both containing three $Zn^{2+}$ ions but with distinct coordination—the first $Zn_3Cys_8His_1$ and the second $Zn_3Cys_9$. This cysteine-rich pre-SET domain makes extensive interactions with the Vefs helical domain and the MCSS domain on one side and with the SET domain on the other. The position and structure of the SET domain in the complex has features that explain the basis of allosteric activation by repressive mark binding to the EED, and suggests a mechanism for oncogenic inhibition.

**Inhibition of PRC2 activity by oncogenic histone H3K27M.** Given that even low level of expression of the histone variant H3K27M is oncogenic and prevents methylation of wild-type histone by PRC2, we determined the structure of human PRC2 with a histone peptide containing the H3K27M substitution. There is unambiguous electron density, in unbiased Fourier maps, for residues 22 to 30 of the peptide (Fig. 2a). Significantly the peptide binds in a canonical manner to the active site of the SET domain, with the Met-27 side chain positioned in the 'lysine' access channel that would be expected to accommodate the target substrate residue. For comparison, Fig. 2b,c show target lysine side chains in the typical SET domain proteins, GLP and Dim5, illustrating the striking spatial similarity to the location of the methionine residue in our structure. There is close agreement in the positions of the Cβ, Cγ, Sδ and Cε of the methionine with the aliphatic counterparts in the lysine side chain. In common with other SET domain structures, the top of the access channel, where the side chain enters, is formed of a series of aromatic residues, which create the hydrophobic environment necessary to accommodate the aliphatic portion of a lysine side chain. At the bottom of the channel, approaching the cofactor methyl donor, is a carbonyl cage conducive to catalysis[22,23]. It is worth noting that the SET domains only methylate the de-protonated form of the target lysine. Thus, the charge properties and configuration of the methionine side chain are similar to a SET-bound lysine.

We established a fluorescence anisotropy assay to measure equilibrium dissociation constants for the binding of histone peptides with either methionine or lysine at position 27 to the PRC2 complex (Fig. 2d). Briefly, the equilibrium dissociation constant for the binding of a carboxy-fluorescein-labelled H3K27M peptide to the PRC2 complex was determined by direct titration (Supplementary Fig. 4a). Equilibrium dissociation constants for unlabelled peptides were then determined using displacement titrations. Peptide binding was strongly cofactor-dependent (Supplementary Fig. 4). Importantly, in the presence of SAM, we observe 16-fold tighter binding of PRC2 towards the oncogenic H3K27M peptide ($K_d = 3.3\,\mu M$) compared with wild-type ($K_d = 52\,\mu M$; Fig. 2e). An important contributory factor to the inhibitory effect of the oncogenic peptide, therefore, derives from its substantially tighter binding.

In their recent analysis, Brown *et al.*[19] quantified the inhibitory affect of a series of peptide constructs varying in the side chain (both natural and synthetic) at position 27. They concluded from these detailed studies that 'the EZH2 active site binds strongly to linear, hydrophobic side chains with little tolerance to extra steric bulk or polar groups'. Our work provides a structural explanation for these observations by revealing that the hydrophobic 'lysine-access' channel readily accommodates a methionine residue in the hydrophobic cavity of this part of the active site. The fact that Norleucine is an even better inhibitor than methionine is also explained by our modelling experiments (Supplementary Fig. 3), which show that this hydrophobic side chain can be well accommodated in this channel. The tighter binding of methioine, and Norleucine, over lysine likely reflects the higher energetic cost of de-solvating the latter.

A significant feature of the histone H3 substrate-binding site in the SET domain of human PRC2 is the well-defined pocket accommodating the Arg-26 residue (Fig. 2g). An arginine residue in the $(-1)$ position is a distinguishing feature of repressive methyl-lysine chromatin marks (Fig. 2f). In all structurally characterized SET domain/repressive substrate complexes, this arginine makes a series of hydrogen bonds to complementary side chains on the SET domain and is a key factor in specific recognition of the target lysine[24–26] (Fig. 2h). The positioning of the H3K27M peptide we observe in the human PRC2 complex structure is therefore fully consistent with that observed in other SET domain structures with repressive histone marks. Here, the position of the guanidinium group of Arg-26 is compatible with hydrogen bonds to the side chains of Asp-652 and Gln-648 of EZH2. The key role that the Arg-26 side chain plays in the recognition of the target sequence by the SET domain is reflected in the 22-fold weaker binding ($K_d = 73\,\mu M$) when the Arg26 is substituted with an Ala (Fig. 2d,e). Thus, by establishing the register of the substrate peptide in the active site, this conserved interaction contributes to the sequence specificity of PRC2. Interestingly, Brown *et al.*[19] have shown that posttranslational modification of residues proximal to the methionine residue reduces effective inhibition by H3K27M. Specifically, asymmetric dimethylation of Arg-26 diminished inhibition approximately 30-fold. In the light of our structure, it seems very likely that the loss of potency of this peptide inhibitor arises from its weaker binding because the two methyl groups cannot be accommodated in the arginine binding pocket. We will discuss the mechanism of the dominant inhibition displayed by H3K27M later.

**Stabilization of the SET domain.** Comparison of the structure of the EZH2 SET domain observed in the complex with that observed in previously determined structures of the isolated domain, reveals unsurprisingly that the most significant differences occur at the interface with other elements of the PRC2 complex (Fig. 3). These are (i) around the conserved GxG loop of SET-N, abutting the cofactor binding site (3b), (ii) the SET-I region, that contributes to the substrate binding site (3c) and (iii) the post-SET region responsible for completing the lysine access channel and active site (3d). (Further analysis of how this is achieved in other SET enzymes in shown in Supplementary Fig. 4).

In human PRC2, the tight turn under the cofactor binding site has the sequence Gly-Trp-Gly (623–625), with the tryptophan side chain oriented away from the SET domain into a hydrophobic pocket composed of residues Val-107 to Met-110 from the EZH2 SAL, Phe-566 to Ser-568 from SUZ12 Vefs, and Leu-616 to Pro-618 from SET-N and Phe-686 from SET-I (Fig. 3b). In the isolated SET domain structure, these important, non-SET-derived interactions are absent.

The EZH2 SET-I region in the human PRC2 complex adopts an anti-clockwise rigid-body rotation relative to its position in isolated structures as viewed in Fig. 3a. Residues 112 to 121 of the

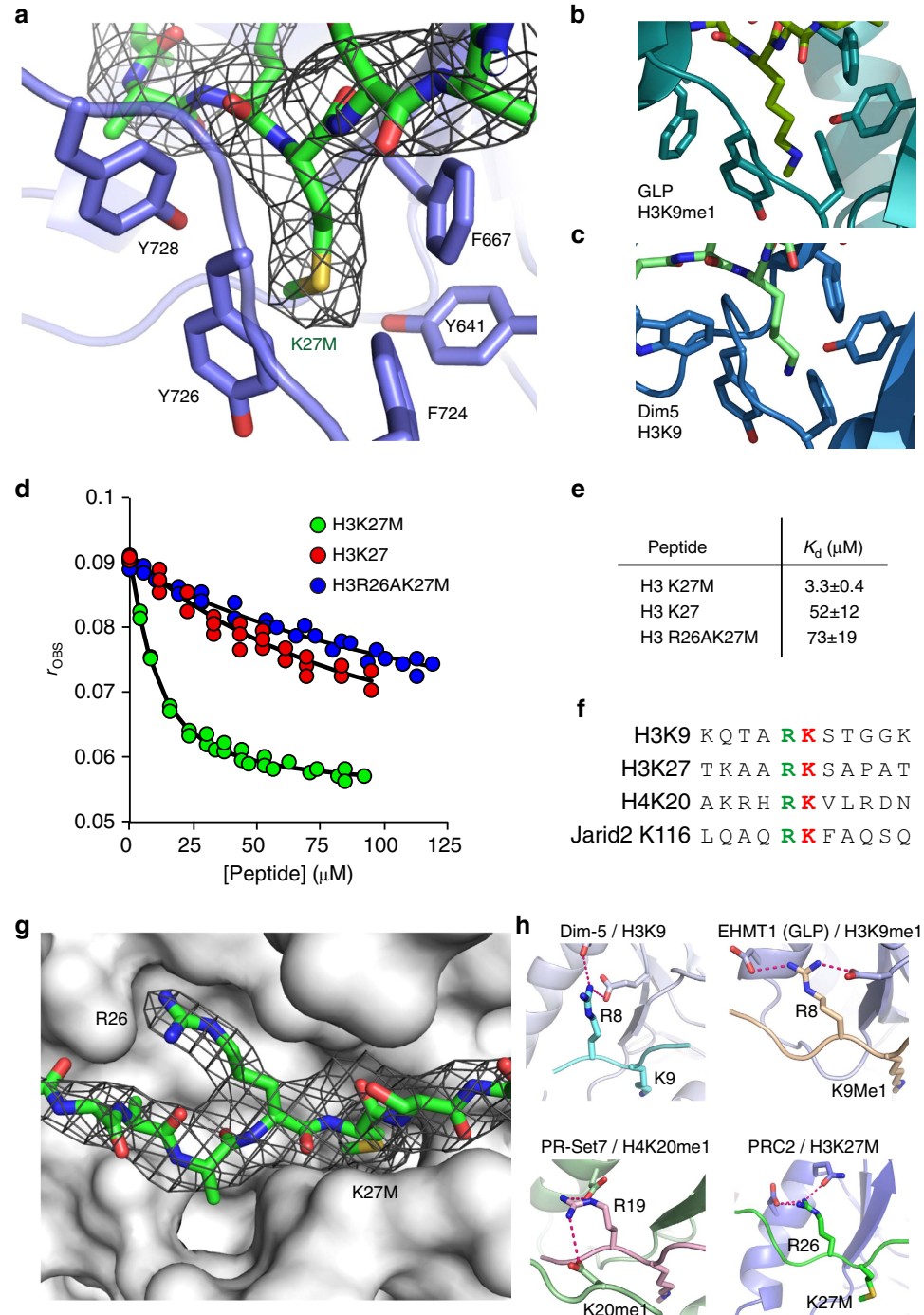

**Figure 2 | Inhibition of PRC2 by H3K27M.** (**a**) Stick representation of the EZH2 'lysine' pocket (blue) with the Fourier Map covering the histone H3K27M side chain (green). An unbiased Fourier Map (Fo-Fc omit map contoured at 2σ) covering the H3K27M peptide (green stick representation) is included. The peptide occupies the canonical substrate binding cleft of EZH2 and there is good density for the methionine side chain. (**b**) Equivalent view of the lysine binding channel in GLP/EHMT1–H3K9me1 (ref. 13) and (**c**) Dim5–H3K9 complexes[26] (lower panel). (**d**) Fluorescence anisotropy displacement titrations in which unlabelled peptides (H3K27M (green circles), H3K27 (red circles) and H3R26AK27M (blue circles)) were added to a mixture of FAM-H3K27M (0.4 μM) and PRC2 (4.3 μM) in the presence of SAM (320 μM). A full description of the binding experiment is provided in the 'Methods' section and in Supplementary Fig. 4 (**e**) Dissociation constants for peptide binding to PRC2. (**f**) Sequence flanking the methylated lysine (red) of repressive chromatin marks indicating the invariant arginine residue (green). (**g**) Surface representation of the substrate binding site in the EZH2 SET domain indicating the Arg(−1) binding pocket and showing the unbiased Fourier Map (Fo-Fc omit map contoured at 2σ). (**h**) Equivalent views of repressive histone substrate binding in the structures of SET domains of PRC2, Dim-5, GLP/EHMT1 and PRSET7 highlighting the presence of related Arg (−1) interactions.

SAL pack against the SET-I region to stabilize its conformation in the active complex (Fig. 3c). In addition to side-chain interactions, this stabilization is mediated by six main-chain hydrogen bonds. Importantly for regulation, in the human PRC2 complex, the conserved acidic residues 584–588 of SUZ12 in turn also pack against residues 112–121 of SAL. Previously, this SUZ12 region

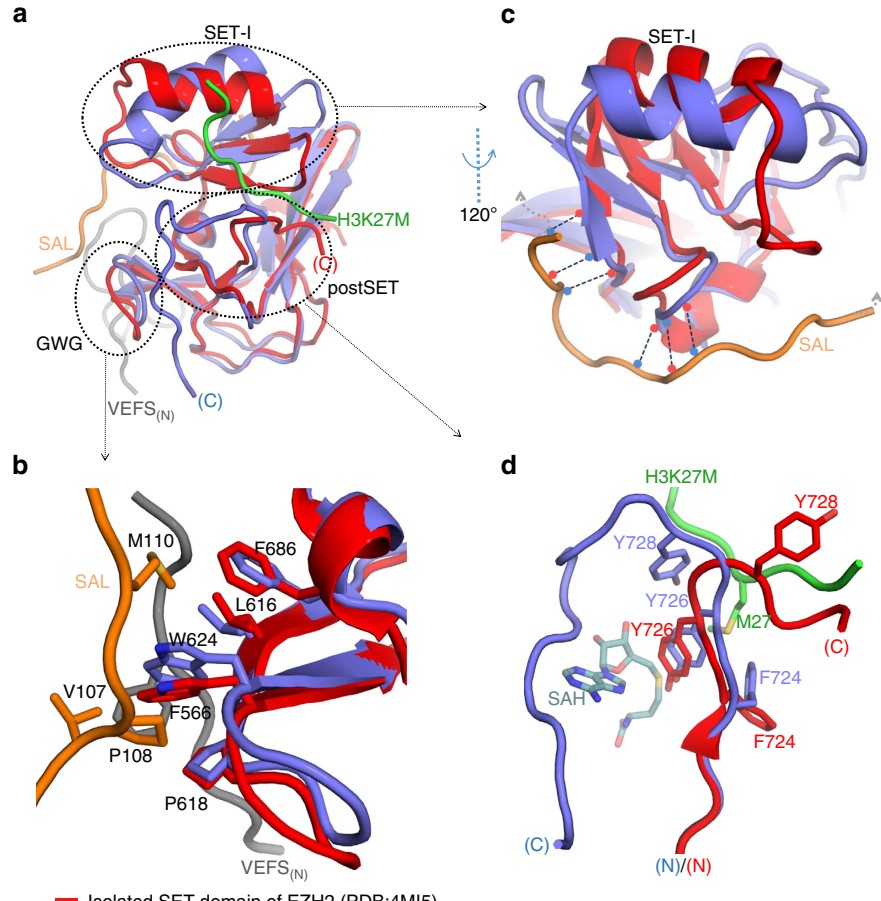

■ Isolated SET domain of EZH2 (PDB:4MI5)

**Figure 3 | Stabilization of the SET-I region.** The active conformation of the SET domain is stabilized by interactions with the Vefs domain and with the SAL and SRM from the N terminus of EZH2. (**a**) Superposition of the human EZH2 SET domain from the PRC2 complex structure (blue) with isolated domain (PDB 4MI5, red). There are three main regions stabilized by interactions in the complex (**b**). The conserved GxG loop is stabilized by interactions with the SAL subdomain from the N terminus of EZH2 (orange) and the region N-terminal to the SUZ12 Vefs domain (grey). (**c**) The SAL subdomain region of EZH2 (orange) also makes a series of main chain interactions with the SET-I region, contributing to the stabilization of the domain and the ordering of the active site. (**d**) There is a significant change in the conformation of the C terminus of the SET domain in the complex to support the completion of the active site.

was shown to mediate binding to histone H3 (31 to 42), resulting in stimulation of PRC2 activity[27]. The structure suggests that this component of PRC2 activation, associated with a dense chromatin environment, occurs by additional stabilization of the SET-I region.

Finally, the structures of the EZH2 post-SET regions, from the isolated and complex forms, differ substantially after residue Tyr-726 (Fig. 3d). In the isolated structure, residues 727–729 partially fold into the substrate-binding channel and thereafter the chain is disordered. In the human complex, the chain from residue 726 takes the opposite trajectory and forms a canonical post-SET structure, contributing residues Tyr-726 and Tyr-728 to the lysine-access channel. We have shown before that cofactor binding is required to stabilize the active conformation of the post-SET domain[25]: this, and the presence of bound peptide in the substrate site, likely contribute to the fully folded active site described here.

Comparison of the structure of the EZH2 SET domain from *C. thermophilum* and human PRC2, reveals that the SET-I region in the former case is significantly displaced outwards, away from the active site (Supplementary Fig. 2c). The arrangement in *C. thermophilum* perhaps reflects the occupancy of the 'lysine' access channel by the bulkier and charged arginine residue.

**Allosteric activation mediated by EED**. There is good electron density to define residues 113–120 of the Jarid2 peptide, bound to EED in the complex (Fig. 4a). It occupies the same site we reported for the isolated EED domain, with the trimethylated lysine residue accommodated in an aromatic cage on the top surface of the β-barrel structure. However in the complex, the Jarid2 peptide makes additional interactions with residues from the SRM moiety of the EZH2 N-domain (Fig. 4b,c). In turn, the SRM helix packs against the SET-I helix (Fig. 4e,f).

The side chain of Arg(−1) (here, Jarid2 R115 equivalent to histone H3R26), makes a salt bridge to acidic residues located in the turn between the SRM extended chain and helix, and two hydrogen bonds with glutamine side chains from the peptide at the (−4) and (+3) positions (Fig. 4b,c). The importance of these interactions was highlighted in earlier studies showing that substitution of Arg(−1) for Ala(−1) led to a peptide that still bound, albeit more weakly, but which no longer allosterically activated PRC2 (ref. 10). Thus, the structure now explains these data, as the salt bridge between the activating peptide and the SRM mediates signalling between the EED binding site and the active site of the SET domain. Interestingly, in the *C. thermophilum* structure, the equivalent Arg(−1) residue is oriented away from the SRM (Fig. 4d), reflecting differences in

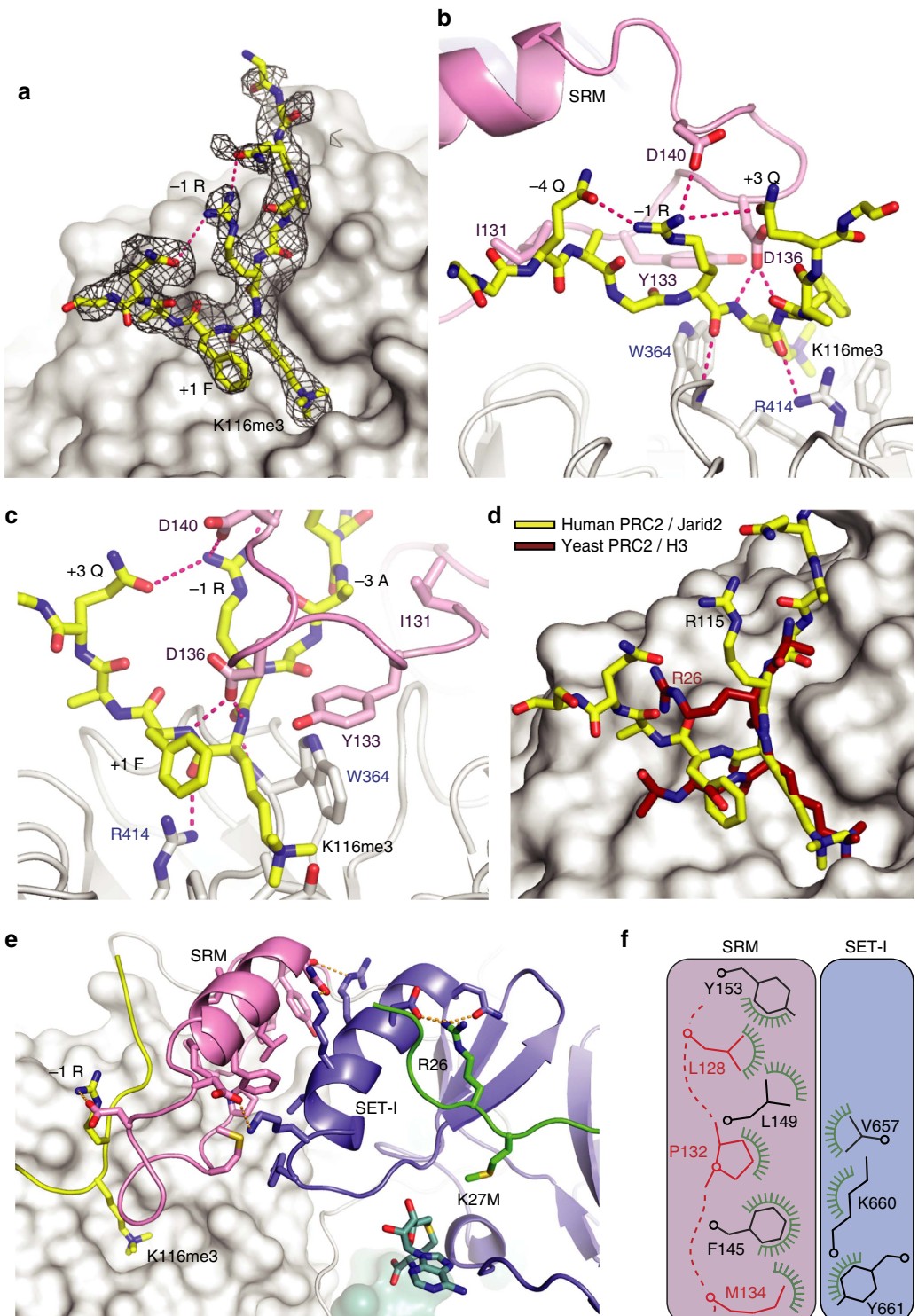

**Figure 4 | Activation of PRC2 by the Jarid2 K116me3 peptide. (a–d)** In this figure, **a,c** and **d** are shown in the same orientation as Fig. 1a, while **b** is an alternative view selected to highlight the interactions of Arg(−1). For clarity, a number of Jarid2 side chains have been omitted. (**a**) Surface representation of EED and unbiased Fourier Map (Fo-Fc contoured at 2σ) showing the binding of the repressive trimethyl peptide from Jarid2 (yellow). The methylated lysine (K116me3) binds in a defined pocket on the surface of EED. (**b**) Stick representation of repressive peptide (yellow) making interactions with EED (white) and with the SRM subdomain of EZH2[N]. The interaction between the conserved Arg(−1) side chain and the SRM is important for positioning of the SRM, which in turn stabilizes the SET-I helix to activate catalysis. (**c**) The interactions between repressive peptide, EED and the SRM. (**d**) Although the Jarid peptide (human structure) and the H3K27me3 peptide (yeast structure) occupy the same binding site in EED, in the yeast structure the crucial Arg(−1) side chain does not bind to the SRM. (**e**) Surface representation of EED, the SRM subdomain (pink) and SET domain (blue) are shown in cartoon representation with key residues as sticks. The Jarid2 peptide (yellow) binds to EED, this stabilizes the SRM, which in turn stabilizes the SET-I facilitating methylation of the H3K27 target (green). (**f**) A schematic showing the hydrophobic interactions between the SRM (helix residues in black, extended chain residues in red) and the SET-I.

the SET-I and SRM, perhaps arising from the presence of the bulkier charged arginine residue in the active site. Asp-136, from the SRM element, makes hydrogen bonds with the peptide main chain amide at the ($+1$) and ($+2$) position (Fig. 4c). The peptide main chain amide and carbonyl at ($-1$) both hydrogen bond to the main chain of residue $364_{(EED)}$ and the carbonyl of the trimethyl lysine residue hydrogen bonds with Arg-$414_{(EED)}$. In addition, the side chains of alanine ($-3$) and phenylalanine ($+1$) sit in shallow hydrophobic pockets on the surface of EED. The alanine at ($-3$) also packs against Ile-131 of the extended strand of SRM (Fig. 4c) providing additional stabilization. Both the Jarid2 and H3K27 peptides, which showed strongest activation of PRC2 (refs 5,10), have an alanine in this position, whereas other repressive peptides showing weaker activation have larger charged side chains.

The stabilization of the SRM by peptide binding to EED leads to activation of EZH2 through the SET-I (Fig. 4e and Supplementary Fig. 6). The interaction between the extended chain and helix of the SRM contains a substantial hydrophobic component contributed by residues including; Leu-128, Pro-132 and Met-134 from the strand and Phe-145, Ile-146, Leu-149, Ile-150 and Tyr-153 from the SRM helix (Fig. 4f). The SET-I helix in turn packs against the SRM helix and contributes three side-chains to the hydrophobic interface just described; Val-$657_{(SET)}$, the aliphatic moiety of Lys-$660_{(SET)}$ and the phenolic ring of Tyr-$661_{(SET)}$. Additionally, Lys-$660_{(SET)}$ makes a salt bridge with Asp-$142_{(SRM)}$ and Lys-$656_{(SET)}$ makes a hydrogen bond with Asn-$152_{(SRM)}$.

Comparison of the structures of *C. thermophilum* PRC2, determined in the presence and absence of EED bound trimethyl lysine peptide, reveals that the extended chain and helix of the SRM are disordered in the absence of peptide. There is very little change in the position of the SET-I helix but the analysis of normalized thermal factors (Supplementary Table 1) shows that it is substantially less mobile when the SRM is ordered upon the binding of the allosteric peptide. This argues that peptide binding to EED, orders the SRM, which in turn stabilizes the structure of the SET-I. Allosteric activation of PRC2 by peptide binding to EED is between 10- and 20-fold. This equates to a stabilization energy of 1.36–1.77 kcal mol$^{-1}$, a rather small energetic difference (less than the strength of a typical protein h-bond). Thus, the observed levels of stimulation of PRC2 activity is consistent with the stabilization of the SET-I structure, and its contribution to the active site of the enzyme, and does not necessitate a substantial conformational change in the SET domain or its binding partners. As in the other SET domains, the PRC2 SET-I region contributes many of the residues that interact with the substrate peptide (Fig. 5a). We were interested in how binding of the substrate peptide was affected by the presence of trimethyl peptide at EED. Binding measurements were performed, as described earlier, in the presence of trimethyl peptide (Fig. 5b). Under these conditions, we observed a 6-fold enhancement of the binding of PRC2 to H3K27M peptide under these conditions, but only a 4-fold enhancement for H3K27 (Fig. 5c). Thus the preference of PRC2 for methionine over lysine at position-27 goes from 16-fold to 22-fold in the presence of repressive marks bound to EED.

## Discussion

Two mechanistically related findings arising from this work are shown schematically in Fig. 6.

First, the PRC2 complex provides a series of interacting segments that stabilize the active form of the SET domain, which is otherwise auto-inhibited when EZH2 is expressed alone. Next, the binding of repressive marks to the EED subunit of the complex further activate the methyltransferase activity of the complex by facilitating the ordering of the SRM element which interacts with, and stabilizes the SET domain. The repressive signal can either come from Jarid2 (ref. 10) or from existing repressed chromatin[5]. The twin aspects of PRC2 product binding to the EED subunit, that is the tighter avidity of the complex for its nucleosomal substrate and allosteric stimulation of activity, together provide an amplification mechanism that enables the PRC2 complex to maintain repressive chromatin states following dilution by DNA replication (Fig. 6).

Second, the oncogenic H3K27M substitution places a methionine side chain into the 'lysine' access channel in the active site. The structure explains the observation that having an unbranched hydrophobic residue at position 27 leads to more effective inhibition, and the importance of the proximal H3 residues, predominantly Arg-26. But how does the lysine to methionine substitution lead to such a pronounced dominant inhibitory effect? Physiological binding of PRC2 to nucleosomes encompasses multiple interactions that although individually weak, in combination, give rise to a high avidity[28]. As in other important biological contexts, the effective avidity is likely to be much lower than the summation of contributing binding energies, and is optimized to achieve the required physiological outcome[29]. In the case of PRC2, the contributing components include SUZ12 binding to H3 (31–42) and RbAp48 binding to H3 (1–10) and H4 helix 1, as well as the SET domain binding its substrate H3K27. Importantly, an additional component of overall avidity is provided by EED binding to H3K27me3. Recent studies in vitro have suggested that K27M mutant nucleosomes inhibit PRC2 at about 2 nM. Here we show that a methionine peptide binds to the PRC2 active site with 22-fold higher affinity than the equivalent lysine peptide in the presence of the SAM cofactor and with a repressive trimethyl lysine peptide bound to EED. In addition to enhancing the avidity of PRC2 for nucleosomes, the lysine to methionine substitution leads to formation of a nonproductive complex because it cannot undergo methyl transfer. Normally, following methylation, the formation of the cofactor product SAH, which binds much weaker, would drive dissociation of the target histone. But, with the methionine substitution, this cannot occur. Thus, we propose that the occurrence of K27M on a small number of histones, in a chromatin dense environment enriched in existing repressive marks, increases the overall avidity of the PRC2/nucleosome interaction such that it effectively sequesters PRC2, preventing further propagation of the repressive mark (Fig. 6). This model is consistent with a recent demonstration that other post-translational modifications on histone tails, that reduce PRC2 avidity for nucleosomes, are able to 'detoxify' the K27M mutation in vivo[19].

## Methods

**Protein expression and purification.** The human PRC2 core was co-expressed in *Escherichia coli* using two expression plasmids, each containing one pair of fragments. Recombinant PRC2 was much less prone to aggregation when EZH2 was expressed in two parts (EZH2$^N$ 1–385 and EZH2$^C$ 421–746), removing the linker between the end of MCSS and the beginning of SANT2, thus 28 residues are missing from the middle lobe as indicated in Fig. 1d. One plasmid, a modified pGex4T1 vector contained EED (residues 77–441) and EZH2 (residues 1–385), such that the EED had an N-terminal glutathione-S-transferase (GST) tag and EZH2 an N-terminal His-tag, each with a thrombin cleavage sequence. Multiple sequence alignments from different species indicated that EZH2 residues 183–210 (inclusive) were not conserved and this region was removed to aid crystallization. Methyltransferase assays[30] showed that constructs without this region were still active. A modified pET49 vector was constructed containing EZH2 (residues 421–746) and SUZ12 (residues 535–685), each with an N-terminal GST tag, with the former containing a Precission 3C cleave sequence and the latter thrombin. Proteins were co-expressed in BL21 (λDE3), immobilized on glutathione resin (GE Healthcare) and separated from tags sequentially, first using alpha thrombin, then with Precission 3C protease. Eluted protein was further purified using ion-exchange chromatography (Source Q, GE Healthcare) and size exclusion

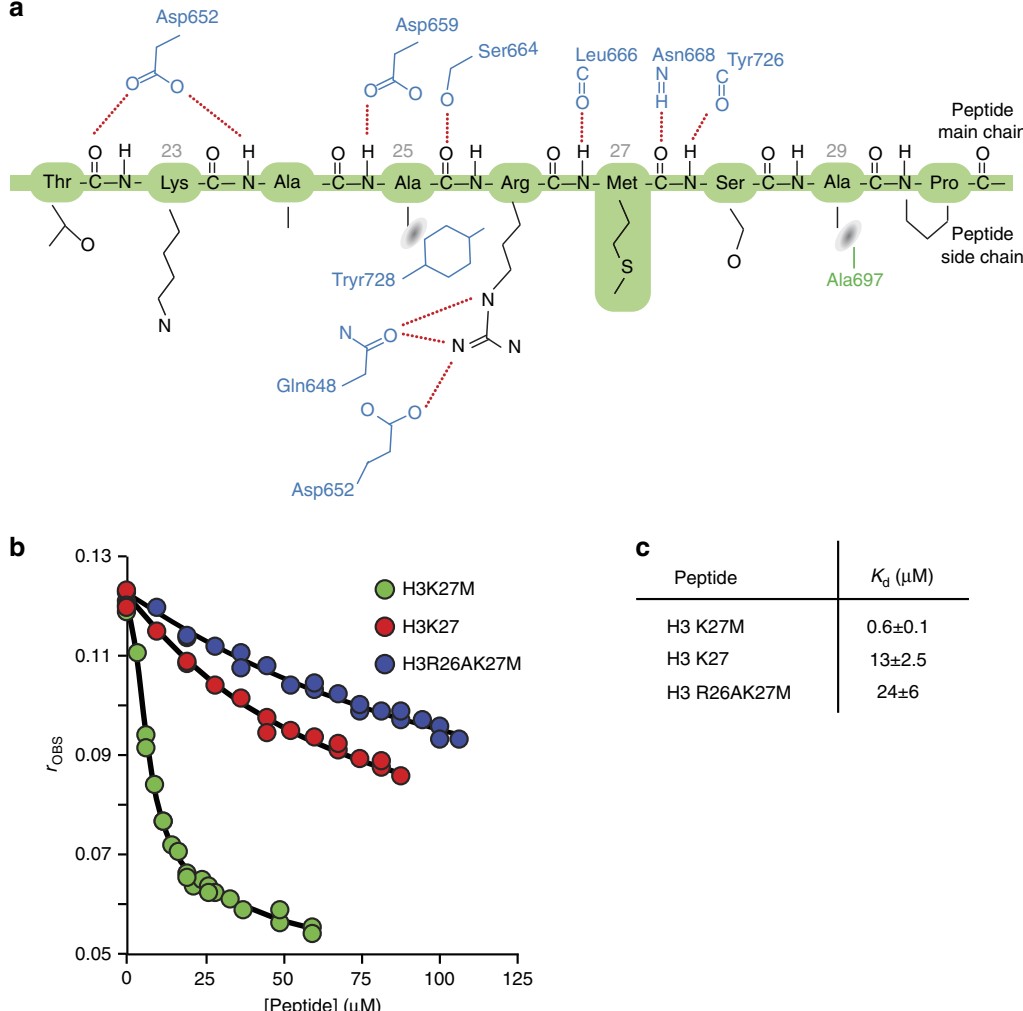

**Figure 5 | Substrate binding is enhanced by activation. (a)** Schematic representation of the principal interactions between the H3K27M peptide and the EZH2 SET domain. The peptide is outlined in green, interactions involving the peptide main chain are shown above the peptide, and with peptide side chains are shown beneath the peptide. For clarity the interactions with the methionine side chain are omitted. SET-I residues are shown in blue and other SET domain residues in green. **(b)** Fluorescence anisotropy displacement titrations in which unlabelled peptides (H3K27M (green circles), H3K27 (red circles) and R26AK27M (blue circles)) were added to a mixture of FAM-H3K27M (0.4 μM) and PRC2 (4.3 μM) in the presence of SAM (320 μM) and H3K27me3 (40 μM). H3K27me3 was used as the activating peptide rather than Jarid116me3. Although both peptides clearly exhibited essentially the same enhancement of binding, at higher substrate peptide concentrations readings in the presence of Jarid116Kme3 were less stable and $K_d$ values were less reliable. A full description of the binding experiment is provided in the 'Methods' section and in Supplementary Fig. 4. **(c)** Dissociation constants for substrate peptides in the presence of repressive peptide.

chromatography (Superdex 200, GE Healthcare) in buffer containing 50 mM Tris pH 8.5, 300 mM NaCl, 1 mM TCEP.

**Crystallography.** Purified PRC2 complex was prepared with H3K27M peptide (H3 residues 22–42), trimethylated K116 Jarid2 peptide (residues 111–121) and S-adenosylmethionine (SAM). The complex was prepared at a protein concentration of 4.8 mg ml$^{-1}$ with 1:10 molar ratio H3K27M peptide, 1:5 molar ratio Jarid2 peptide and 1:10 molar ratio SAM. The crystals were grown at 18 °C using the vapour diffusion technique in MRC sitting drop plates (Hampton Research). The drops were prepared by mixing equal volumes of PRC2 complex with a reservoir solution containing 400 mM ammonium citrate pH 6.5 and 25% PEG3350. The crystals were transferred into reservoir solution containing 20% ethylene glycol before flash cooling in liquid nitrogen. The diffraction data were collected at Diamond Light Source on beamline IO2 at wavelengths 0.980 and 1.283 Å (Selenium and Zinc edges). The data were integrated using XDS[31] and scaled with SCALA. The phases were generated in PHASER[32] using a combination of molecular replacement (using the structures of EED (3IIW) and EZH2 SET domain (4MI5) as search models) with zinc anomalous scattering. Followed by iterative rounds of building with the electron

density being improved by non-crystallographic symmetry averaging using the CCP4 programs PARROT and DM[33]. The initial molecular replacement found several solutions for both EED and the SET domain. A number of these were tested as single combined search models; for one of these, the molecular replacement using the combined EED/SET search model then found four copies of this pairing within an asymmetric unit that had a sensible packing arrangement. Confidence in this solution was increased as clear density was visible for the EED binding helix of EZH2 and also density could be seen for the Vefs domain helices. Further improvement of the maps was achieved at this stage by combining phases from the latest model with phases from the zinc anomalous data such that eventually most residues in PRC2 could be added. The positions of the zinc atoms were identified from the zinc anomalous data using the program PHENIX[34]. The zinc anomalous data only gave low resolution phase information but it did help at this stage of building. Also, the zinc sites determined only using the anomalous data, confirmed the general correctness of the partial molecular replacement solution. During the course of building the human PRC2 complex, the yeast co-ordinates became available[20]. The areas with adjacent extended chains from several subunits, which were initially difficult to assign, for example in the SAL/SET interface region, benefitted from reference to this structure. Standard refinement was performed with REFMAC5 (ref. 35) and CNS[36] was

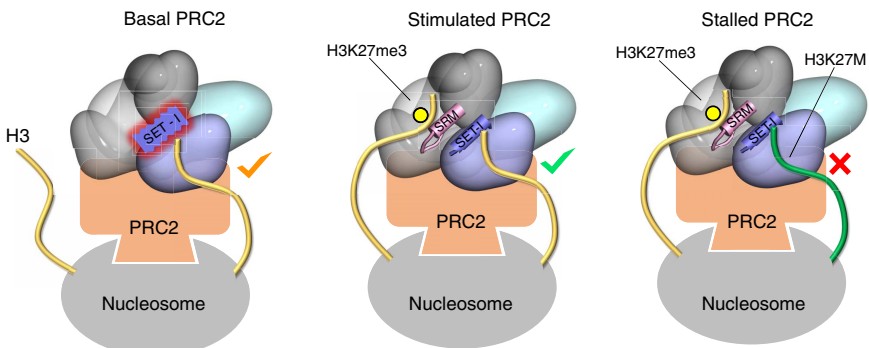

**Figure 6 | Overall model.** The basal activity of PRC2 towards H3K27 (left) is activated by the binding of a 'repressive peptide' to EED (middle), mediated by stabilization of SRM and thus the SET-I. The figure indicates that in the absence of these interactions, the SET-I has more thermal mobility. The propagation of repressive chromatin is mediated by activation through EED. When oncogenic histone mutant (H3K27M) binds to the SET active site (right), the additional contribution to avidity and absence of turnover leads to 'stalling' of PRC2, preventing propagation of the H3K27me3 mark. In addition to the H3 tail interactions with the minimal complex discussed above, the extended PRC2 complex (orange shape) makes extensive ncRNA[38], and protein and DNA interactions with the nucleosome (grey shape).

used to generate composite omit maps. Manual model building was carried out in Coot[37]. The figures were created with Pymol (The PyMOL Molecular Graphics System, Version 1.7.4 Schrödinger, LLC).

**Binding measurements.** Equilibrium dissociation constants for the interaction of PRC2 (P) with FAM-H3K27M peptide (L) to form the complex PRC2:FAM–K27M (PL) were determined using anisotropy titrations. For any mixture of PRC2 and FAM-H3K27M, the observed anisotropy ($r_{OBS}$) is given by:

$$r_{OBS} = \frac{\alpha r_{PL}[PL] + r_L[L]}{\alpha[PL] + [L]} \tag{1}$$

where $r_{PL}$ and $r_L$ are the anisotropies of the complex and the free peptide, and $\alpha$ ($= 1$ in this case) is the fluorescence intensity of the complex divided by that of the free peptide. The data were analysed using nonlinear least-squares fits to equation (1) with $r_{PL}$, $r_L$ and the dissociation constant ($K_{d,L}$) as variables, and PL and L calculated in the usual way:

$$[PL] = \frac{(K_{d,L} + [P_0] + [L_0]) - \sqrt{(K_{d,L} + [P_0] + [L_0])^2 - 4[P_0][L_0]}}{2} \quad \text{and}$$

$$[L] = [L_0] - [PL]$$

where the subscript 0 indicates the total concentration of the species.

Equilibrium dissociation constants for the interaction of PRC2 (P) with the unlabelled peptides (K27M and R26AK27M) were determined using a displacement assay in which aliquots of the unlabelled peptide (N) were added to a preformed PRC2:FAM–K27M complex. In the case of unlabelled K27M methylation could be a factor. We therefore prepared separate solutions of the pre-formed complex for each point of the titration. We monitored anisotropy over time and noted that the reading remained stable for at least 60 s following addition of the K27. The anisotropy reported was taken within 15 s and we are therefore confident that methylation is not affecting the data. The data were analysed using nonlinear least-squares fits to equation (1) with PL and L calculated as follows:

The free protein concentration ([P]) is the root of the cubic equation:

$$C_3[P]^3 + C_2[P]^2 + C_1[P] + C_0 = 0 \tag{2}$$

where

$C_3 = 1$
$C_2 = -[P_0] + K_{d,L} + K_{d,N} + [L_0] + [N_0]$
$C_1 = -[P_0]K_{d,L} - [P_0]K_{d,N} + K_{d,L}K_{d,N} + [N_0]K_{d,L}$
$C_0 = -[P_0]K_{d,L}K_{d,N}$

where $K_{d,N}$ is the equilibrium dissociation constant for binding of N to PRC2 and [N_0] is the total concentration of N. PL and L are then calculated using:

$$[PL] = \frac{[L_0][P]}{K_{d,L} + [P]} \quad \text{and} \quad [L] = [L_0] - [P] \tag{3}$$

All anisotropy titrations were performed at 20 °C using a Jasco FP-8500 fluorimeter. The buffer (25 mM Hepes (pH 7.25), 25 mM NaCl, 1 mM TCEP, 0.01% Brij-35) contained SAM, SAH or SAM plus trimethylated K27 peptide as required.

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

## Acknowledgements

We thank Xiaoli Xiong for advice on structure determination, Evangelos Christodoulou for assistance with cloning and Roksana Ogrodowicz for assistance with crystallization. The peptides used in this study were provided by the Crick Peptide Chemistry service. This work was supported by the Francis Crick Institute which receives its core funding from Cancer Research UK, the UK Medical Research Council and the Wellcome Trust. We greatly acknowledge Diamond Light Source for access to synchrotron time under proposal MX9826.

## Authors contributions

All the authors performed the experiments and contributed to the writing of the manuscript.

## Additional information

**Accession Code:** The structure is deposited with the PDB under accession code 5HYN.

**Competing financial interests:** The authors declare no competing financial interests.

