## [Peer Review File · Nature Communications]

Reviewer #1 (Remarks to the Author):

The authors have addressed my concerns and I am happy to recommend publication in Nature Communications.

Reviewer #2 (Remarks to the Author):

The authors present a revised version of their manuscript reporting the crystal structure of the core of human PRC2 in complex with a Jarid2 peptide containing trimethylated K112 and a histone H3 peptide containing the K27M mutation. The authors have satisfactorily addressed the points raised by this reviewer and the new data strengthen the conclusions of the manuscript. I strongly support publication of this version of the manuscript in Nature Communications.

Reviewer #3 (Remarks to the Author): In this manuscript, Gamblin and coworkers report the crystal structure of the human PRC2 complex containing EZH2, EED, and the VEFS fragment of SUZ12 bound to Jarid2K116me3 and the H3K27M peptides. In addition, fluorescent anisotropy displacement assays were performed to investigate binding of the H3K27 substrate and H3K27M oncogenic mutant to the PRC2 complex. The inclusion of the binding data strengthens the manuscript in comparison to the original version. This work offers insights into the structure and allosteric regulation of the human PRC2 complex and complements the previously reported structure of the *Chaetomium thermophilum* PRC2. The following comments should be addressed before considering this manuscript for publication: 1) In the displacement assays with PRC2, SAM, and the unmodified H3K27 peptide, measurements were made within 15 seconds to minimize methylation of the peptide, which would complicate determination of the dissociation constant. However, given the high concentration of SAM (320 μ M), PRC2 (4.3 μ M), and H3K27 peptide (0 ~ 90 μ M) used in the assay, it is conceivable that some methylated H3K27 peptide product is formed within 15 seconds of initiating the assay. The authors should perform controls to verify whether product is formed within the 15 seconds, and if so, quantify how much is formed. Alternatively, control displacement assays with the H3K27 peptide should be performed using a non-reactive SAM analogue, such as sinefungin, to substantiate the binding data obtained using SAM.

Reviewer# 1 (Response to reviewer): Thank you for the positive comment.

Reviewer# 2 (Response to reviewer): Thank you for the positive comment.

Reviewer# 3 (Response to reviewer): We were aware of the potential for methylation of substrate peptide to complicate this measurement. It is for this reason that we took the precaution of first assessing the stability of the anisotropy measurement over the timescale of measurement and then selecting a sufficiently short time point so that we could rule out methylations as an issue. To clarify this point in the manuscript we have added the following text to the methods (page 15) H3K27me3 was used as the activating peptide rather than Jarid116me3. Although both peptides clearly exhibited essentially the same enhancement of binding, at higher substrate peptide concentrations readings in the presence of Jarid116Kme3 were less stable and Kd values were less reliable.

Our initial approach was to use JaridK116me3 in these measurements. However, we encountered an issue with stability of the readings at a high substrate peptide concentrations. Therefore, in order to generate a reliable set of Kd measurements we have reported the binding constants measured using the H3K27me3 peptide to activate the complex. We have added a note to the appropriate figure legend to clarify this issue. In the case of unlabelled K27 methylation could be a factor. We therefore prepared separate solutions of the pre-formed complex for each point of the titration. We monitored anisotropy over time and noted that the reading remained stable for at least 60 sec following addition of the K27. The anisotropy reported was taken within 15 seconds and we are therefore confident that methylation is not affecting the data.

We acknowledge that the meaning of the term 'repressive' is context dependent, but feel that it is used appropriately. For additional clarity we have added this sentence to the introduction (page 2). Thus H3K27me3 functions as a repressive mark in terms of gene expression but has an activating capacity with respect to PRC2 function.